# Effects of Rumen Fermentation Characteristics on Stress-Related Hormones and Behavior in Sheep

**DOI:** 10.3390/ani13233701

**Published:** 2023-11-29

**Authors:** Sanggun Roh, Da-Hye Kim, Tetsuro Shishido, Shin-Ichiro Ogura

**Affiliations:** 1Laboratory of Grassland-Animal Production and Ecology, Graduate School of Agriculture Science, Tohoku University, Osaki 9896711, Japan; qianrige.t8@dc.tohoku.ac.jp (Q.);; 2Laboratory of Animal Physiology Science, Graduate School of Agriculture Science, Tohoku University, Sendai 9808572, Japan; sanggun.roh.a3@tohoku.ac.jp; 3Division of Animal Genetics and Bioinformatics, National Institute of Animal Science, Rural Development Administration, Wanju 55365, Republic of Korea; dhkim0724@korea.kr

**Keywords:** stress response, cortisol, growth hormone, open field test, rumen volatile fatty acid, sheep

## Abstract

**Simple Summary:**

The intricate relationship between dietary intake and behavioral manifestations in ruminants, specifically sheep, constitutes the focal point of this investigation. The endeavor was embarked upon with an objective to discern the impact of dietary variations, particularly those rich in concentrates, on the behavioral and hormonal responses of sheep under stress-inducing scenarios. The findings elucidate a noteworthy propensity for escape attempts in sheep subjected to a high-concentrate diet when placed in novel environments, juxtaposed with their counterparts on a control diet. Furthermore, the dietary regimen manifested a tangible influence on the equilibrium of specific volatile fatty acids within the rumen, which ostensibly correlates with their behavioral responses. While the underlying mechanisms that govern these observations remain enshrouded in complexity, the revelations of this study underscore the imperative of meticulous dietary selection and management in livestock. Through the astute understanding and optimization of animal dietary practices, a conduit is forged towards the enhancement of animal welfare and the refinement of farming practices, thereby conferring tangible benefits upon the animals and broader society.

**Abstract:**

This study investigated the relationship between rumen fermentation, stress-related hormones, and behavior in sheep with the aim of providing insights for improving animal welfare and feed management practices. Eight lambs were assigned to either a high concentration or control group. Blood samples were collected for hormone analysis, and an open field test was conducted to observe behavioral stress responses. The results showed that diet composition may affect the behavior of ruminants in response to stressors and novel situations, as evidenced by the higher number of escape attempts in the high-concentration group. In addition, analyses of individual volatile fatty acids (VFAs) showed a significant positive correlation between the acetic acid/propionic acid ratio and sniffing behavior of the novel object (*p* < 0.05, ρ = −0.414). These findings have important implications for animal welfare and feed management practices. Overall, this study provides insights into the potential impact of diet composition on the behavior of ruminants in response to stressors and novel situations, highlighting the importance of improving animal welfare through feed management practices. Further research is needed to fully elucidate the mechanisms underlying the complex relationship between rumen fermentation, stress-related hormones, and behavior in ruminant animals.

## 1. Introduction

Ruminant feed design has traditionally been based on nutrient balance, feed intake, palatability, digestion in the rumen, and the effects of the feed on ruminant metabolism, immunity, and productivity [1,2,3]. However, previous studies have directly or indirectly shown that changes in rumen fermentation status in ruminants can lead to changes in hormone levels, including hormones related to animal temperament and behavioral stress response, such as growth hormone (GH) and cortisol. For example, saturated fatty acids inhibit the release of adrenocorticotropic hormone (ACTH) in the anterior pituitary cells of rats [4]. In a study by Matsunaga et al., ruminal infusion of VFAs strongly inhibited the decrease in plasma concentration of GH concentration in sheep [5]. These studies suggest that rumen fermentation characteristics play an important role in the stress-related hormone concentrations of ruminant animals. 

Changes in the endocrine system can profoundly affect the stress responses and temperaments of animals. For instance, GH has been shown to affect animal temperament, with increased blood GH concentrations correlating with escalated aggressive behavior in fish [6]. This relationship between GH and temperament has also been observed in ruminants: mithuns with high plasma GH concentrations have been noted to exhibit aggressive behaviors [7]. Additionally, genetic polymorphisms in GH may influence stress responses through GH concentration in steers [8]. These findings highlight the importance of further exploration of the complex relationship in rumen fermentation, particularly the roles of VFAs, endocrine responses, and stress-related behaviors in ruminants. However, the exact mechanisms underlying rumen–endocrine–stress response behavior remains to be fully elucidated.

Therefore, this study aimed to investigate the effect of VFAs on stress-related hormone concentrations and behavior in sheep under different dietary conditions (high concentrate vs. control). Blood samples were collected for hormone analysis, and an open field test was conducted to observe the behavior of the lambs. This study will contribute to a better understanding of the complex relationship between rumen fermentation, stress-related hormones, and behavior in ruminant animals and may provide insights for improving animal welfare and feed management practices.

## 2. Materials and Methods

### 2.1. Animals and Treatments

This study was conducted in 2016 at the Kawatabi Field Science Center, Graduate School of Agricultural Science, Tohoku University (Osaki, Miyagi, Japan). Eight Suffolk lambs, each at the age of 8 months with an average body weight of 36.2 ± 5.3 kg, were used at the start of the experiment. Six lambs were castrated males and two were females. Lambs were housed in individual pens measuring 1.8 × 1.5 m with straw bedding. The lambs were divided into two groups *(n* = 4 per group): the high concentrate group (H group), wherein the diet was a roughage–concentrate ratio of 20:80 on a dry matter basis, and the control group (C group), wherein the diet was orchardgrass silage and alfalfa hay cubes (Table 1). The groups were counterbalanced for sex and body weight. The total digestible nutrient (TDN) intake was balanced between the two treatment groups, with each lamb receiving 0.88 kg/day according to the Japanese Feeding Standard for Sheep [9]. The lambs were fed twice per day at 9:00 AM and 5:00 PM and were provided with water and mineral mixture ad libitum. 

### 2.2. Experimental Design

The study was conducted utilizing a repeated-measures design, wherein the same lambs were subjected to the same treatment across different periods. This design was strategically chosen for several pivotal reasons. Firstly, it allowed for the detailed observation and analysis of the impact of the feed combination on sheep behavior and hormone levels at different time points. Secondly, considering the availability of sheep and limitations of experimental resources, the repeated-measures design facilitated the extraction of more information from a smaller sample size, enhancing the efficiency and cost-effectiveness of the experiment. Thirdly, conducting multiple measurements on the same group of sheep minimized the impact of individual differences on the experimental results, ensuring more accurate and reliable data.

Moreover, the repeated-measures design was also chosen to mitigate potential habituation effects in the Open Field Test (OFT). Given that animals might alter their behavioral responses upon repeated exposures to the same test environment and stimuli, maintaining the same treatment for each animal throughout the study aimed to keep any habituation effects consistent across all testing points. This is particularly pertinent as, in a crossover design, each animal would undergo the OFT under each treatment condition, potentially introducing variability in the results due to differential habituation effects.

Given the lack of prior research on the impact of feed design on both physiological and behavioral stress responses in sheep, it was challenging to determine an appropriate acclimatization period for a crossover design. The repeated-measures design, therefore, mitigated potential carryover effects that might have arisen from insufficient washout periods between treatments in a crossover design.

### 2.3. Sampling and Measurements

The experimental period was divided into two phases: a transition period lasting 8–10 d and a sampling period lasting 4 d. This procedure was repeated twice under identical conditions. On day 1 of each sampling period, 10 mL of blood was collected from each lamb via jugular venipuncture 2 h after the morning feed. This timing was consistent across different days and aligned with the time points for both the behavioral experiment and rumen fluid collection. Our objective was to assess the impact of dietary treatment on hormone levels, thus necessitating a uniform sampling schedule. It is important to note that blood sampling was not conducted following the behavioral test. This decision was made to ensure that the sampling times were consistent and focused on the effects of the treatment rather than the immediate aftermath of the behavioral test. The samples were centrifuged at 2000× *g* for 15 min at 4 °C, and the resulting plasma was collected and stored at −80 °C until analysis of cortisol and GH concentrations was performed. Plasma cortisol concentration was measured using the DetectX^®^ Cortisol Enzyme Immunoassay Kit (K003-H1/H5; Arbor Assays, Ann Arbor, MI, USA). Plasma GH was measured using a time-resolved fluoroimmunoassay (TRIFMA), first introduced by Løvendahl et al. [10] in 2003 for ruminant animals, to test Growth Hormone (GH) levels. This assay utilizes monoclonal antibodies that are raised against recombinant bovine GH. One of these antibodies is immobilized on the wells of a microtiter plate, while the other is labeled with Europium (Eu^3+^), serving as a tracer. The assay exhibits a sensitivity of 0.1 ng/mL and operates within a working range of 0.2 to 200 ng/mL. Given the structural similarity and nearly identical amino acid sequences among ovine, caprine, and bovine GH forms, a close cross-reactivity is anticipated. This methodological approach allows for a precise and reliable measurement of GH levels in these species.

On day 2 of each sampling period, 100 mL of ruminal fluid was collected from each individual using oral stomach tubing 2 h after the morning feeding. The decision to sample during this window was strategic, aiming to capture the most pronounced effects of the feeding treatment on the ruminal environment and, subsequently, on stress-related hormones and behavior, given that VFAs production is correlated with stress hormone secretion. The collected rumen fluid samples were filtered through double-layered medical gauze and placed in an icebox for storage until centrifugation. After all samples were collected, they were centrifuged at 10,000× *g* for 4 min to remove solid particles. The samples were then stored at −80 °C until further analysis. The concentrations of VFAs (acetic acid, propionic acid, and butyric acid) in the rumen fluid samples were measured using high-performance liquid chromatography (Tohoku Electronic Industrial, Sendai, Japan). The HPLC analysis was completed by Rooke et al. in 2014, and detailed experimental procedures and methods were introduced in a related publication [11]. For each run, 20 µL samples was injected twice onto a Bio-Rad HPX 87H ion-exchange column (Bio-Rad Laboratories Ltd, Watford, UK), with dimensions of 300 mm by 7.8 mm, safeguarded by a 30 mm cation H+ ion-exchange resin guard column. The mobile phase consisted of 0.005 M sulfuric acid, with the system maintained at a temperature of 50 °C and a flow rate set at 0.8 mL per minute. 

### 2.4. Behavioral Test

The Open Field Test (OFT) was used to assess behavioral responses to a novel object and social isolation. During day 4 of each sampling period, each lamb was individually placed in a novel arena measuring 5 × 5 m. Each session was conducted 2 h after the morning feed. The observation period began when the hind legs of each animal entered the open field. During the measurement period, the behavior in the arena was recorded for 20 min using two video cameras set on a wooden panel surrounding the arena. The behavioral types recorded were as follows: ‘latency to sniff the novel object for the first time’, which measures the time it takes for a sheep to initiate contact with the novel object using its nose; ‘number of bleats’, accounting for both audible vocalizations with mouth open or closed; ‘sniffing of the novel object’, indicating the count of instances where the sheep’s nose made contact with the object; ‘sniffing of the ground or wall’, reflecting the instances of nasal contact with the ground or arena walls; and ‘escape attempts’, describing the actions of sheep attempting to jump or collide with the walls in an effort to leave the open field (Table 2). After each test, the floor of the arena was cleaned to ensure consistent testing conditions.

To provide further clarity on the timing and sequence of the behavioral tests in relation to other sampling procedures, it is important to note that the Open Field Test was conducted on day 4 of each experimental period. This scheduling allowed for a gap day between the ruminal fluid collection on day 2 and the behavioral tests, ensuring that the sheep were calm and unstressed by previous procedures. The timing of the behavioral tests was aligned with the other sampling events, each conducted 2 h post-feeding.

### 2.5. Statistical Analysis

In this study, two-way analysis of variance (ANOVA) was performed to confirm the effects of treatment and period on ruminal VFA, blood GH, and cortisol concentrations. After normalizing the distribution using log transformation, two-way ANOVA was used to analyze the behavioral variables in the OFT. The model included treatment, period, and treatment-by-period interactions as factors. To examine the relationships between the VFAs and behavioral variables in the OFT, we performed a Spearman correlation analysis. All statistical analyses were performed using R version 4.2.3 and environment for statistical computing. R version 4.2.3, such as ‘car’ for two-way ANOVA and “stats” (cor.test function) for Spearman’s correlation, were used.

## 3. Results

### 3.1. VFAs in the Rumen and Plasma Hormones

Total VFA in the rumen and plasma hormones were not significantly affected by treatment, period, or the interaction between the two (Table 3). However, the proportion of acetic acid was significantly higher in group C (71.5%) than in group H (67.8%; *p* = 0.016). In addition, a significant difference was found in butyric acid levels, which were markedly lower in group C (7.6%) than in group H (13.6%; *p* = 0.002). Meanwhile, an interaction effect was observed between the treatment and period on the proportion of acetic acid (*p* = 0.018). 

In terms of plasma hormone concentrations, no significant effects of the treatment or their potential interactions were observed (Figure 1). Nevertheless, in the H group, the postprandial plasma GH concentration tended to exceed that of the C group. Specifically, the plasma GH concentrations for the C and H groups were 1.22 ng/mL (SE = 0.07 ng/mL) and 1.91 ng/mL (SE = 0.54 ng/mL), respectively, in the first period and 1.25 ng/mL (SE = 0.14 ng/mL) and 1.98 ng/mL (SE = 0.40 ng/mL), respectively, in the second period. Moreover, a significant effect of the period was noted for cortisol, with lower concentrations observed in period 2 than in period 1 (*p* < 0.05).

### 3.2. OFT

In the OFT, the H group demonstrated a higher frequency of escape attempts (17 times) than the C group (3 times; *p* < 0.05), as shown in Figure 2. However, no significant differences were found between the two groups in other behavioral metrics within the OFT. Although this experiment aimed to observe differences in behavioral stress responses under different dietary levels, a significant effect of the period was also observed in our two-way ANOVA. Specifically, the number of bleats was significantly higher in period 2 than in period 1 (*p* < 0.05), and the number of escape attempts was significantly lower in period 2 than in period 1 (*p* < 0.05).

### 3.3. Relationship between VFA and Behaviors

Table 4 presents the significant correlations between VFAs and behavioral variables in the OFT. A positive correlation was found between the acetic acid/propionic acid ratio and the frequency of sniffing the novel object (ρ = 0.545, *p* < 0.05).

## 4. Discussion

In this study, we utilized a repeated-measures design to ensure consistency in treatment across all exposures to the OFT, which was crucial for controlling habituation effects. Habituation could influence the sheep’s behavioral responses and introduce variability in our results. By keeping the treatment constant, we aimed to minimize this variability. 

Our findings revealed that treatment, period, or the interaction between them had no significant effects on total VFA and acetic acid/propionic acid in the rumen. This suggests that the overall rumen fermentation status remained relatively stable across different dietary conditions and periods. However, significant differences were observed in the proportions of acetic and butyric acids. The proportion of acetic acid was significantly higher in group C than in group H. This may be attributed to the different feed compositions of the two groups, with group C receiving a diet of orchardgrass silage and alfalfa hay cubes, which favors the production of acetic acid in the rumen. In some studies, roughage, such as hay, tended to produce a higher concentration of ruminal acetic acid than concentrate feeds [12]. In contrast, the proportion of butyric acid was markedly lower in group C than in group H, and this may be due to the possibility that the high concentration of the diet in group H may have promoted butyric acid production. Although the period had no influence on the concentration of VFAs, an interaction effect between the treatment and period was detected for the proportion of acetic acid. This indicates that the effect of dietary treatment on the proportion of acetic acid was not consistent across different periods, suggesting that other factors, such as the adaptation of the rumen microbial composition to the diet over time, played a role. Plasma concentrations of hormones, particularly GH, tended to be higher in group H than in group C. However, no significant differences were observed in other aspects owing to diet treatment, period, or the interaction between the two. In addition to these findings, some studies on the effects of VFA mixtures on GH suggest that an increase in ruminal VFA inhibits the secretion of GH. For instance, intravenous infusion of acetate was found to suppress the secretion of GH in sheep [5]. In another study involving sheep, acetate, propionate, and butyrate were administered separately via intravenous infusion, and the results showed that butyrate suppressed the elevated plasma GH concentration due to the presence of GH stimulants, whereas acetate did not have this effect [13]. 

In the context of the dietary treatments implemented in this study, it is pivotal to underscore that the two feed treatments were designed under the same TDN provision but were differentiated by their concentrate-to-forage ratio. Consequently, the high-concentrate feed group (H) inevitably exhibited a lower dry matter intake compared to the forage feed group (C). This discrepancy in intake between the two groups instigated a notable variation in the degree of rumen fill, which, based on the aforementioned studies regarding the influence of rumen fill on GH secretion, could potentially emerge as a significant factor influencing GH secretion, aside from the VFA concentration.

Other studies have demonstrated that manipulating the rumen fill, such as by filling the rumen with a water balloon, decreased the blood GH concentration [14]. Therefore, considering the lower dry matter intake and consequently, the lower rumen fill in group H, this might have contributed to the higher GH levels observed in this group. Thus, in future studies on the impact of ruminal VFA on GH, the rumen fill factor, including variations caused by different dietary compositions, should be given greater consideration. Meanwhile, the number of escape attempts in the OFT was significantly higher in group H than in group C. The H group also tended to have higher plasma GH concentrations than the C group, which is consistent with previous studies reporting that higher GH concentrations are associated with stronger behavioral stress responses and aggressiveness in ruminant animals [7,8]. However, in our analysis, a significant positive correlation was not observed between escape attempts and plasma GH concentration. Future studies with larger sample sizes are required to further investigate this relationship. Meanwhile, significant changes were observed in the plasma cortisol concentrations, bleats, and escape attempts between the two periods. The reduced cortisol levels could be attributed to the increased familiarity of the sheep with the overall experimental conditions at the time of the second blood collection. Adaptation to an experimental environment, including handling and blood collection procedures, can reduce stress responses, thereby lowering cortisol levels. Additionally, the increased bleeding and decreased escape attempts during the OFT in the second period may further support this interpretation, suggesting that the sheep became more comfortable with the testing environment. The higher number of bleats suggests that the sheep were less anxious and more interactive, whereas the reduced number of escape attempts could signify a decreased desire to flee from a familiar setting. The significant positive correlation between the acetic acid/propionic acid ratio and the sniffing behavior of the novel object suggests that the rumen fermentation characteristics of individual lambs may be related to their behavior in the OFT. Even if the total VFAs in the rumen did not change in our study, a decrease in the acetic acid/propionic acid ratio can lead to a decrease in rumen pH [15]. This decrease in rumen pH can result in reduced feed intake in dairy cows [16]. Feeding motivation has been reported to influence locomotor and exploratory behaviors in animals [17]. Therefore, the lower acetic acid/propionic acid ratio observed in some lambs may have led to lower feeding motivation or reduced appetite and a subsequent decrease in exploratory behavior in ruminant animals. Given these observations, it is evident that further research is necessary to delineate the causal relationships implicated by these findings. Future investigations should consider larger sample sizes and employ varied methodological approaches to robustly confirm or elucidate the causal mechanisms between diet composition, rumen fermentation characteristics, and stress-related behaviors, with particular attention to the role of the acetic acid/propionic acid ratio. Additionally, the impact of dietary composition, particularly the forage to concentrate ratio (F:C), on ruminal microbial populations and their subsequent influence on stress responses merits consideration. The acetate to propionate (A/P) ratio serves as a pivotal indicator of the ruminal environment’s adaptation to variations in F:C. Research involving Angus cows has demonstrated that changes in the F:C ratio significantly affect the A/P ratio, ruminal pH, and the composition of ruminal microbiota [18]. These alterations in microbial populations are potentially pivotal in modulating the physiological and behavioral responses of ruminants under stress. This underscores the complex interplay between diet, ruminal fermentation, and animal behavior, and emphasizes the need for further exploration into the multifaceted interactions among these factors.

Despite the absence of a direct examination of rumen microbiota in our study, the potential role they may play is intriguing. The gut–brain axis, a well-documented phenomenon in monogastric animals, illustrates a significant influence of gut microbiota on brain function and behavior. For instance, alterations in gut microbiota composition in patients with schizophrenia have been linked to cognitive dysfunctions, potentially involving the microbial biosynthesis of neurotransmitter precursors such as tyrosine [19]. Additionally, the abundance of specific bacteria like Bifidobacterium and Lactobacillus has been associated with stress responses and mood disorders in humans, with lower counts observed in individuals with major depressive disorder (MDD) [20]. In a similar vein, Bifidobacterium longum has been shown to normalize anxiety-like behavior in mice, a process that might engage vagal pathways [21]. While such interactions have yet to be investigated in ruminants, these findings suggest that rumen microbiota could potentially exert analogous effects through pathways that have not yet been identified. Future research should aim to explore the potential influence of rumen microbiota on stress-related behaviors and physiological responses in ruminants. Investigating these relationships would enhance our understanding of the rumen–brain axis and could carry significant implications for the welfare and management of these animals.

## 5. Conclusions

In conclusion, our study suggests that the composition of the diet and the resulting rumen environment play a significant role in modulating stress-related hormonal responses and behaviors in ruminants. While the mechanisms underlying these effects are complex and warrant further study, our data indicate that lambs fed a higher concentrate diet exhibited more escape tendencies, and a lower acetic acid to propionic acid ratio correlated with reduced curiosity. Thus, even without overt ruminal fermentation disorders, high concentrate feeding regimes could potentially impose chronic and subtle stress on livestock. This highlights the importance of considering the impact of diet composition on animal welfare in livestock management practices.

## Figures and Tables

**Figure 1 animals-13-03701-f001:**
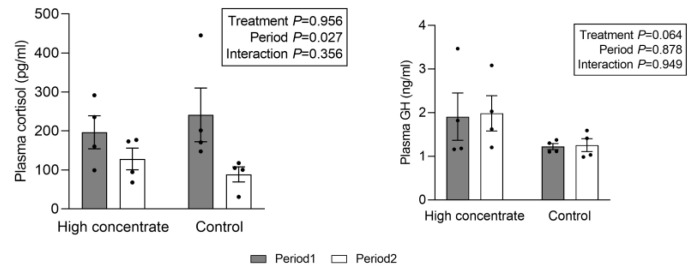
Effect of different treatments on the plasma concentration of hormones 2 h post-feeding. Data are expressed as the mean ± standard error of the mean (SEM). Filled bars represent period 1, and open bars represent period 2. Two-way ANOVA after log transformation (*n* = 8 per treatment).

**Figure 2 animals-13-03701-f002:**
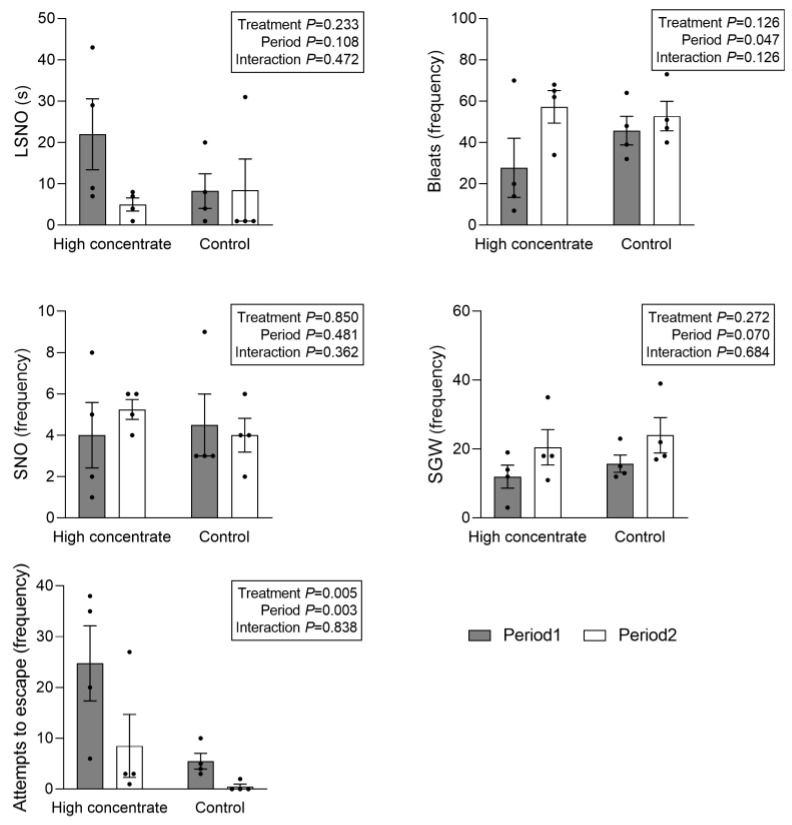
Effect of different treatments on behaviors measured in the OFT. Data are expressed as the mean ± SEM. Filled bars represent period 1, and open bars represent period 2. Two-way ANOVA after log transformation (*n* = 8 per treatment). LSNO represents latency to sniff the novel object for the first time, bleats represent the number of vocalizations emitted by the sheep during the test period, SNO represents sniffing of the novel object, SGW represents sniffing of ground or wall, and escape attempts in this context refer to instances wherein the test sheep attempted to escape.

**Table 1 animals-13-03701-t001:** Chemical composition of concentrate diet and composition and nutritive value of two experimental feeds in the high concentrate and control groups.

**Ingredient (g/kg of DM)**		**High Concentrate**	**Control**
Orchardgrass silage		80	400
Alfalfa hay cube		120	600
Concentrate		800	0
**Chemical Composition (g/kg of DM)**	**Concentrate**	**High Concentrate**	**Control**
Crude protein	195	170	146
Total digestible nutrient	845	787	557
Ether extract	38	38	30
Neutral detergent fiber	204	271	536
Acid detergent fiber	105	160	384
Ash	76	84	117

**Table 2 animals-13-03701-t002:** Ethogram of sheep behaviors observed during the open field test.

Behavior	Description
Latency to sniff novel Object	Time taken for a sheep to initiate contact with the novel object using its nose.
Number of bleats	Count of audible vocalizations with mouth open or closed.
Sniffing novel object	Instances where the sheep’s nose made contact with the novel object.
Sniffing of ground or wall	Instances of nasal contact with the ground or arena walls.
Escape attempts	Actions of sheep attempting to jump or collide with the walls in an effort to leave the open field.

**Table 3 animals-13-03701-t003:** Effect of different treatments on the concentration of total VFAs and proportion of each VFA in the rumen of lambs 2 h after feeding. Two-way ANOVA after log transformation (*n* = 8 per treatment).

	Period	High Concentrate		Control		*p* Value		
		Mean	SD	Mean	SD	Treatment	Period	Treatment × Period
Total VFA (mM)	1	45.3	14.8	44.6	11.8	0.443	0.213	0.393
	2	48.3	11.3	60.0	17.2			
Acetic acid (%)	1	65.7	1.3	73.0	2.7	0.016	0.744	0.018
	2	69.8	1.9	69.9	1.9			
Propionic acid (%)	1	19.2	1.8	20.9	2.0	0.119	0.742	0.742
	2	18.2	3.9	20.9	2.8			
Butyric acid (%)	1	15.1	0.8	6.0	1.5	0.002	0.991	0.053
	2	12.0	4.8	9.2	3.0			
Acetic acid/propionic acid	1	3.5	0.2	3.5	0.2	0.368	0.528	0.286
	2	4.0	0.4	3.4	0.2			

**Table 4 animals-13-03701-t004:** Spearman correlation of rumen VFA levels with open-field behaviors. * *p* < 0.05, *n* = 16. LSNO represents latency to sniff the novel object for the first time, bleats represent the number of vocalizations emitted by the sheep during the test period, SNO represents sniffing of the novel object, SGW represents sniffing of ground or wall, and escape attempts in this context refer to instances wherein the test sheep attempted to escape the open field by jumping and colliding against the wall.

Variables	Volatile Fatty Acids				
	Acetic Acid	Propionic Acid	Butyric Acid	Total VFA	Acetic Acid/Propionic Acid
Open-field behavior					
LSNO (s)	0.145	0.241	0.094	0.183	−0.016
Bleats (frequency)	0.232	0.388	−0.162	0.265	−0.274
SNO (frequency)	−0.095	0.241	0.210	−0.085	0.545 *
SGW (frequency)	−0.251	−0.179	−0.236	−0.244	−0.128
Escape attempts (frequency)	−0.207	−0.216	0.322	−0.195	0.178

## Data Availability

Data supporting the findings of this study are available from the corresponding author upon request.

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
