# Peer review of "Effects of Rumen Fermentation Characteristics on Stress-Related Hormones and Behavior in Sheep"

_animals, 2023, doi:10.3390/ani13233701_

Round 1

Reviewer 1 Report

Comments and Suggestions for Authors

This study is commendable for investigating the interplay between rumen fermentation, stress-related hormones, and behavior in sheep, which has implications for animal welfare and feed management. However, there are some aspects that could be further refined. The sample size of eight lambs may be considered relatively small, potentially limiting the generalizability of the findings. Additionally, while the study highlights correlations between variables, causality is not established, indicating a need for more in-depth research. The language used in presenting the results could be more precise; for instance, stating the specific values of the acetic acid/propionic acid ratio and sniffing behavior correlation would enhance clarity. Furthermore, it would be beneficial to explore additional factors that could contribute to the observed behaviors and hormonal responses, providing a more comprehensive understanding. In addition, the following points need further attention on the part of the authors.  Firstly, in Table 1, it is recommended to avoid using abbreviations to enhance clarity for readers. Moreover, if abbreviations are used in other tables, they should be appropriately defined for better comprehension. Regarding the statistical analysis, there seems to be a discrepancy as the authors mention both two-way ANOVA and one-way ANOVA. This inconsistency needs to be addressed and clarified in order to ensure the accuracy of the results. Additionally, the adequacy of the provided data for a comprehensive paper may be a concern, and further data collection or analysis may be necessary. The reference section is notably sparse, indicating that a more extensive literature review could enhance the study's context and relevance. A more thorough exploration of relevant prior research would strengthen the study's foundation. Overall, while the study offers valuable insights, addressing these points would refine the presentation and bolster the robustness of the findings.

Author Response

Dear Reviewer,

Thank you for your insightful comments and suggestions, which have greatly helped in enhancing the quality of our manuscript. We have thoroughly revised the manuscript based on your feedback and have detailed our responses to each of your comments below. The changes made in response to your comments are highlighted in yellow and gray.

1
Dear Reviewer,
Thank you for your insightful comments and suggestions, which have greatly helped in enhancing 
the quality of our manuscript. We have thoroughly revised the manuscript based on your feedback 
and have detailed our responses to each of your comments below. The changes made in response to 
your comments are highlighted in yellow and gray.
Comment 1: “The sample size of eight lambs may be considered relatively small, potentially 
limiting the generalizability of the findings.”
Response 1: We appreciate the reviewer’s point regarding our sample size. It is important to note 
that while striving for robust statistical power, we also placed a high priority on the welfare of the 
animals involved in our study. The invasive nature of the procedures, such as the collection of rumen 
fluid via orogastric tubing and blood sampling from the jugular vein, necessitated a careful balance 
between achieving meaningful statistical results and minimizing pain and distress to the animals. 
We intentionally designed the experiment to keep the sample size as small as possible without 
compromising the scientific validity, adhering to the principles of Replacement, Reduction, and 
Refinement (the 3Rs) in animal research. We acknowledge the limitations that this ethical stance 
imposes on the generalizability of our findings, and we have discussed these in the manuscript (p. 
8, line 279 and 300). Future studies with larger sample sizes could potentially confirm and extend 
our observations, provided that the welfare of the animals can be assured.
Comment 2: While the study highlights correlations between variables, causality is not established, 
indicating a need for more in-depth research.
Response 2: Thank you for your comment regarding the distinction between correlation and 
causality. Our study design, which incorporated dietary interventions, was intended to explore 
causative effects. However, we understand that establishing causality in biological systems often 
requires a more longitudinal approach. We have revised the discussion section to emphasize the 
correlational nature of our findings and the need for further studies to investigate causality more 
definitively (p. 8, line 298).
Comment 3: The language used in presenting the results could be more precise; for instance, stating 
the specific values of the acetic acid/propionic acid ratio and sniffing behavior correlation would 
enhance clarity.
Response 3: Thank you for your comments regarding the precision of language used to describe the 
statistical findings in our manuscript. We have taken great care to ensure that our presentation of the 
results is both accurate and clear. Specifically, in the results section, we have provided the exact 
value of the Spearman's correlation coefficient (ρ=0.545) along with the significance level (P<0.05) 
to reflect the relationship between the acetic acid/propionic acid ratio and the sniffing behavior 
observed during the Open Field Test. These details are directly stated in the text and further 
highlighted with an asterisk in Table 3 to emphasize their significance.
We believe this approach adequately conveys the strength and significance of the correlations to 
the readers, allowing for a transparent and straightforward interpretation of the data. If there are any 
additional aspects of the language precision that could be enhanced further, we welcome your 
suggestions and are open to making any necessary adjustments to improve the clarity of the 
2
manuscript.
Comment 4: Furthermore, it would be beneficial to explore additional factors that could contribute 
to the observed behaviors and hormonal responses, providing a more comprehensive understanding.
Response 4: In response to your suggestion, we have included a discussion on the potential role of 
rumen microbiota in modulating stress-related behaviors and physiological responses in ruminants
(p. 8, line 304). Although our study did not directly assess the rumen microbiota, the literature from 
monogastric animal models provides compelling evidence that warrants such consideration in future 
ruminant research.
Comment 5: Firstly, in Table 1, it is recommended to avoid using abbreviations to enhance clarity 
for readers.
Response 5: Thank you for your suggestion regarding the use of abbreviations in Table 1. In 
accordance with your recommendation, we have revised the table to replace abbreviations with their 
full terms to enhance clarity for our readers.
Comment 6: Moreover, if abbreviations are used in other tables, they should be appropriately 
defined for better comprehension.
Response 6: Following your recommendation, we have now detailed the full descriptions of the 
open field behaviors of table3. Additionally, we have provided explicit definitions for each of these 
behavioral responses in the Methods section of our paper to ensure clear understanding for our 
readers (p. 4, line 154).
Comment 7: Regarding the statistical analysis, there seems to be a discrepancy as the authors 
mention both two-way ANOVA and one-way ANOVA. This inconsistency needs to be addressed 
and clarified in order to ensure the accuracy of the results.
Response 7: Thank you for pointing out the inconsistency in our description of the statistical 
analyses. We have carefully reviewed our manuscript and would like to confirm that all statistical 
analyses were indeed conducted using two-way ANOVA. Any mention of one-way ANOVA was an 
error and has now been corrected to reflect the accurate method employed (p. 4, line 170. We 
appreciate your attention to detail, which has helped us improve the precision of our manuscript.
Comment 8: Additionally, the adequacy of the provided data for a comprehensive paper may be a 
concern, and further data collection or analysis may be necessary.
Response 8: We appreciate your observation on the adequacy of the data presented in our manuscript. 
We recognize the importance of comprehensive data to support our findings and conclusions. The 
dataset reported was meticulously collected and analyzed to ensure the robustness of our study 
within the ethical constraints of animal research. We have strived to present a balanced account by 
discussing the limitations and context of our data collection in the manuscript. In the study, we 
adhered to the principles of the 3Rs (Replacement, Reduction, and Refinement) to minimize animal 
use and ensure ethical treatment. This ethical stance inevitably influences the sample size but is a 
critical consideration in the field of animal research. We believe our dataset provides valuable 
insights into the interplay between rumen fermentation characteristics, stress-related hormones, and 
behavior in sheep, serving as a preliminary exploration of this complex field. We have now 
3
expanded the Discussion section to further elaborate on the potential implications of our findings 
and to call for additional studies that could augment our dataset with a larger sample size or 
longitudinal analyses, which would be invaluable in confirming and extending the patterns observed.
Comment 9: The reference section is notably sparse, indicating that a more extensive literature 
review could enhance the study's context and relevance. A more thorough exploration of relevant 
prior research would strengthen the study's foundation.
Response 9: Thank you for your constructive feedback on the reference section of our manuscript. 
In response to your suggestion, we have diligently expanded our literature review to provide a more 
comprehensive context for our study. We agree that a robust foundation of previous research is 
essential to frame our study's contributions effectively. With the newly added citations, we have 
endeavored to encapsulate a broader spectrum of related work, thereby situating our preliminary 
investigation within the wider scientific discourse. We believe that these additions not only 
strengthen the study’s foundation but also underscore the novelty of our research as an initial 
exploration into the dietary modulation of rumen fermentation and its potential effects on stressrelated behaviors in ruminants.

Sincerely,

Author

Reviewer 2 Report

Comments and Suggestions for Authors

Dear Authors,

The manuscript entitle ""Effects of rumen fermentation characteristics on stress-related hormones and behavior in sheep" can be improved.

Please find below my suggestions and doubts for its improvement.

With kind regards,

Reviewer

Author Response

Dear Reviewer,

Thank you for your insightful comments and suggestions, which have greatly helped in enhancing the quality of our manuscript. We have thoroughly revised the manuscript based on your feedback and have detailed our responses to each of your comments below. The changes made in response to your comments are highlighted in blue and gray.

Dear Reviewer,
Thank you for your insightful comments and suggestions, which have greatly helped in enhancing 
the quality of our manuscript. We have thoroughly revised the manuscript based on your feedback 
and have detailed our responses to each of your comments below. The changes made in response to 
your comments are highlighted in blue and gray. 
Comment 1: In the study, there was a possible effect due to the habituation of the animals. Could 
you please explain if the same people were involved during the experiments? Because animals seem 
to be receptive to people and habits.
Response 1: Thank you for raising a valid point concerning the potential impact of human 
interaction on the habituation of animals in our study. We understand the importance of controlling 
for such variables to ensure the integrity of our behavioral observations. Throughout the study, from 
the transition period leading up to the experiments until the final day, we implemented stringent 
measures to minimize the sheep's contact with humans. The same individual was responsible for the 
routine care and sampling processes to maintain consistency in human-animal interactions. For 
procedures such as blood collection and rumen fluid sampling, an additional randomly selected 
person assisted the primary caretaker. Importantly, during the Open Field Test (OFT), we employed 
video recording to monitor the behavior of the sheep without any human presence in their line of 
sight. This approach was specifically chosen to avoid any potential influence that the presence of a
person might have on the animals' behavior, ensuring that their responses were as natural and 
undisturbed as possible.
Comment 2: -L39= keyword: ruminants? Could be added to know in which specie was this study 
performed even if it seems obvious due to the rumen fatty volatile acids.
Response 2: Thank you for your suggestion to include more specific keywords in our manuscript. 
We agree that clarity is essential for readers to understand the context and focus of our study. As 
recommended, we have now added 'sheep' to the list of keywords to explicitly indicate the species 
used in our research.
Comment 3: L58= could you please expand a bit explaining what you mean with “strong 
temperament” and how do you define it? Which characteristics are involved in an animal of this 
specie to consider it with“strong temperament”.
Response 3: We appreciate your request for clarification on the term “strong temperament” as it 
relates to our study. Upon reviewing our references and considering the precision of terminology, 
we have updated our manuscript to replace the term "strong temperament" with "aggressive 
behaviors."
Comment 4: -L75Could you please add the age average of the lambs.
Response 4: As per your suggestion, we have updated our manuscript to specify that the average 
age of the lambs at the time of the study was 8 months. This information has been added to the 
Materials and Methods section, Line 75.
Comment 5: Table 1: This part goes at the bottom outside part of the table. DM, dry matter; CP, 
crude protein; 109 TDN, total digestible nutrient; EE, ether extract; NDF, neutral detergent fibre; 
ADF, acid detergent fiber.
Response 5: Thank you for your valuable comment on the use of abbreviations in Table 1. In our 
effort to enhance the readability and clarity of our manuscript, we have revised Table 1 by replacing 
all abbreviations with their corresponding full terms. This change has been made to ensure that all 
readers can easily understand the table content without referring to the abbreviations at the bottom 
or elsewhere in the text.
Comment 6: L120=please detail the methodology of the time-resolved fluoroimmunoassay.
Response 6: Thank you for your valuable comment. We have updated our manuscript to specify that
detail of TRIFMA methodology.
Comment 7: L131= please detail the methodology or at least cite from which protocol did you 
follow in order to perform this analysis.
Response 7: Thank you for your valuable comment. We have updated our manuscript to specify that 
detail of HPLC methodology.
Comment 8: L143= what do you mean with this paragraph? I am assuming is the ethical approval, 
should this be in here? Or placed somewhere else. It does not seem to follow up what it is written 
in this part, probably at the animals and treatments sections or at the “Institutional Review Board 
Statement:”
Response 8: Thank you for your valuable feedback. I apologize for any confusion caused. I have 
removed the redundant sections as per your suggestion.
Comment 9: L145= 2.4. Statistical analysis should be “2.5”, numbering is repeated.
Response 9: Thank you for pointing out the numbering error in the manuscript. I have corrected it 
as per your suggestion. I appreciate your attention to detail in helping improve the quality of the 
manuscript.
Comment 10: L170 and L174.Figure 1 and 2 = please italicize the “P” value. For the representation 
of the graphs, could you please add a small square.
Response 10: Thank you for your valuable comment. We have updated our manuscript as per your 
suggestion. 
Comment 11: Table 3= please remove the space between “escape attempts”.
Response 11: Thank you for your valuable comment. We have updated our manuscript as per your 
suggestion. 
Comment 12: L209= probably this word “instrumental” is not the adequate, could you please find 
a synonym or more appropriate word.
Response 12: Thank you for your suggestion. We have replaced “instrumental” with "pivotal" as 
per your recommendation.
Comment 13: L209-221= This paragraph could be shortened and be explained more precisely, since 
you are speaking about habituation in the following part of the discussion.
Response: Thank you for your valuable feedback. We have revised the mentioned paragraph to make 
it more succinct.
Comment 13: L211= “Open Field Test (OFT)” this OFT was already defined before, there is no 
need to re do it again.
Response 13: Thank you for your valuable feedback. We have updated our manuscript as per your 
suggestion.
Comment 14: L222 and L279, L285=please write instead d 2A/P” “Acetic/Propionic”.
Thank you for your valuable feedback. We have updated our manuscript as per your 
recommendation.
Comment 15: L241= please cite “some studies” which citation.
Response 15: Thank you for pointing out the need for citation regarding the studies mentioned. We 
would like to clarify that the reference to 'some studies' is immediately followed by an illustrative 
example, 'For instance, the intravenous infusion of acetate was found to suppress the secretion of 
GH in sheep,' which is duly cited [5].
Comment 16: L249=this Total Digestible Nutrients (TDN) was already defined before, please just 
leave TDN.
Response16: Thank you for your valuable feedback. We have updated our manuscript as per your 
recommendation.
Comment 17: L281= Even if the total VFAs in the rumen do not change (where? In your study? Or 
elsewhere…because if it is in your study, it should be “did not change in our study” …
Response17: Thank you for your valuable feedback. We have updated our manuscript as per your 
recommendation.
Comment 18: L289= could you please conclude which diet do you think it could help lambs to 
suffer less stress. he higher concentrate or the lower one according to your investigation.
Response 18: Thank you for your constructive feedback on our conclusion. We have revised it to 
more directly address the impact of different diets on stress in lambs. Despite the complexity of 
dietary effects on stress response mechanisms, our observations lead us to cautiously suggest that 
high-concentrate diets, even when not causing apparent ruminal disorders, may contribute to chronic 
stress in livestock. We have incorporated this perspective into our conclusion, highlighting the 
importance of optimizing feed composition to enhance animal welfare.

Sincerely,

Author
